# nAdder: A scale-space approach for the 3D analysis of neuronal traces

**Minh Son Phan**[1,2], **Katherine Matho**[3,4], **Emmanuel Beaurepaire**[1], **Jean Livet**[3], **Anatole Chessel**[1]*

**1** Laboratory for Optics and Biosciences, CNRS, INSERM, Ecole Polytechnique, IP Paris, Palaiseau, France, **2** Institut Pasteur, Université de Paris Cité, Image Analysis Hub,Paris, France, **3** Sorbonne Université, INSERM, CNRS, Institut de la Vision, Paris, France, **4** Cold Spring Harbor Laboratory, Cold Spring Harbor, New York, United States of America

* anatole.chessel@polytechnique.edu

## Abstract

Tridimensional microscopy and algorithms for automated segmentation and tracing are revolutionizing neuroscience through the generation of growing libraries of neuron reconstructions. Innovative computational methods are needed to analyze these neuronal traces. In particular, means to characterize the geometric properties of traced neurites along their trajectory have been lacking. Here, we propose a local tridimensional (3D) scale metric derived from differential geometry, measuring for each point of a curve the characteristic length where it is fully 3D as opposed to being embedded in a 2D plane or 1D line. The larger this metric is and the more complex the local 3D loops and turns of the curve are. Available through the GeNePy3D open-source Python quantitative geometry library (https://genepy3d.gitlab.io), this approach termed nAdder offers new means of describing and comparing axonal and dendritic arbors. We validate this metric on simulated and real traces. By reanalysing a published zebrafish larva whole brain dataset, we show its ability to characterize different population of commissural axons, distinguish afferent connections to a target region and differentiate portions of axons and dendrites according to their behavior, shedding new light on the stereotypical nature of neurites' local geometry.

**Data Availability Statement:** No data was generated as part of this study. We actually actively reused publicly available published data, as described section 3.6. The code is available as a

## Auhor summary

To study how brain circuits are formed and function, one can extract neuron traces, i.e. the precise path that neuron arbors take in the brain to connect to other neurons. New techniques enable to do so with increasingly higher throughput, up to every single neuron with so called 'connectomic' approaches. Up to now, the geometry of those traces has not been a focus of study and has mainly been analysed in bulk/on average. Here, we propose to quantitatively analyse the local 3D geometry of the curves that comprise neuron arbors. We introduce an algorithm that determines whether a locally-defined curve is best fit to a line, a plane or a 3D structure. We use it to compute a single number at each point of the trace, termed local 3D scale, that measures the characteristic size of the local 3D structure: the larger this local 3D scale metric, the more the neuron's curve meanders in 3D locally.

python library, described here: https://genepy3d.gitlab.io/.

**Funding:** EB received funding from Agence Nationale de la Recherche (https://anr.fr/) under contract ANR-11-EQPX-0029 Morphoscope2 and ANR-10-INBS-04 France BioImaging JL received funding from Fondation pour la Recherche Médicale (https://www.frm.org/) (DBI20141231328) and Agence Nationale de la Recherche (https://anr.fr/) under contracts LabEx LIFESENSES (ANR-10-LABX-65) and IHU FOReSIGHT (ANR-18-IAHU-01) EB and JL received funding from European Research Council (Horizon 2020 programme, grant No 951330 HOPE) The funders had no role in study design, data collection and analysis, decision to publish, or preparation of the manuscript.

**Competing interests:** The authors have declared that no competing interests exist.

We reanalyse published neuronal traces to demonstrate that our local geometry approach enables to better characterize a neuron's morphology, with direct relevance to understanding its development and function. The local 3D scale metric will be useful in all neuroscience research that works with neuronal traces, bringing a new, geometric layer of information.

## 1 Introduction

Throughout the history of neuroscience, the analysis of single neuron morphologies has played a major role in the classification of neuron types and the study of their function and development. The NeuroMorpho.Org database [1, 2], which collects and indexes neuronal tracing data, currently hosts more than one hundred thousand arbors of diverse neurons from various animal species. Technological advances in large-scale electron [3–6] and fluorescence microscopy [7–11] facilitate the exploration of increasingly large volumes of brain tissue with ever improving resolution and contrast. These tridimensional (3D) imaging approaches are giving rise to a variety of model-centered trace sharing efforts such as the MouseLight (http://mouselight.janelia.org/, [8]), Zebrafish brain atlas (https://fishatlas.neuro.mpg.de/, [12]) and drosophila connectome projects (https://neuprint.janelia.org/, [13]). Crucially, the coming of age of computer vision through advances in deep learning is now offering ways to automate the extraction of neurite traces [14–16], a process both extremely tedious and time consuming when performed manually. This is currently resulting in a considerable increase in the amount of 3D neuron reconstructions from diverse species, brain regions, developmental stages and experimental conditions, holding the key to address multiple neuroscience questions [17]. Methods from quantitative and computational geometry play a major role in handling and analyzing this growing body of data in its full 3D complexity, a requisite to efficiently and accurately linking neuronal anatomy with other properties such as function, development, and pathological or experimental alteration. Furthermore, morphological information is of crucial importance to address the issue of neuronal cell type classification, in addition to molecular data [18, 19].

An array of geometric algorithmic methods and associated software has already been developed to process neuronal reconstructions [20–23]. Morphological features enabling the construction of neuron ontologies, described for instance by the Petilla convention [24], have been used for machine learning-based automated neuronal classification [25]. These features have also been exploited to address more targeted questions, such as comparing neuronal arbors across different experimental conditions in order to study the mechanisms controlling their geometry [26]. Morphological measurements are also employed for an expanding range of purposes in the context of large-scale tracing efforts based on sparse fluorescent labeling or dense microscale connectomic reconstructions with 3D electron microscopy, e.g. to proofread reconstructions [27], probe changes in neuronal structure and connectivity during development [28] or identify novel neuron subtypes [10].

So far however, metrics classically used to study neuronal traces tend to rely on elementary parameters such as length, direction and branching; as such, they do not enable to finely characterize and analyze neurite trajectories. One particularly interesting parameter to fill that gap is their local geometrical complexity, i.e. whether they adopt a straight or convoluted path at a given point along their trajectory. This parameter is of particular relevance for circuit studies, as it is tightly linked to axons' and dendrites' development and their role in information processing: indeed, axons typically follow simple paths within tracts while adopting a more

complex structure at the level of terminal arbor branches that form synapses. Moreover, the sculpting of axonal arbors by branch elimination during circuit maturation can result in convoluted paths [29, 30]. A neurite segment thus provides information on both its function and developmental history. Available metrics such as tortuosity (the ratio of curvilinear to Euclidean distance along the path of a neurite) are generally global and average out local characteristics; moreover, traces from different neuronal types, brain regions or species can span vastly different volumes and exhibit curvature motifs over a variety of scales, making it difficult to choose which scale is most relevant for the analysis. One would therefore benefit from a generic method enabling to analyze the geometrical complexity of neuronal trajectories 1) locally, i.e. at each point of a trace, and 2) across a range of scales rather than at a single, arbitrary scale.

Methods based on differential geometry have been very successfully applied in pattern recognition and classic computer vision [31, 32]. In particular, the concept of scale-space has led to thorough theoretical developments and rich practical applications. The key idea is that starting from an original signal (an image, a curve, a time series, etc.), one can derive a family of related signals that estimate the original one as viewed at various spatial or temporal scales. This enables to select and focus the analysis on a specific scale of interest, or to remove noise or a low frequency background signal; in addition, scale-space analysis also provides multiscale descriptions [33]. On 2D curves, the mean curvature motion is a well-defined and broadly applied scale-space computation algorithm [32]. So far, however, comparatively little has been proposed concerning 3D curves, such data being less common [34]. Applications of curvature and torsion scale-space analyses for 3D curves have been reported [35] but remain few and preliminary; the inherent mathematical difficulty represents another reason for this gap. Neuronal trace analysis clearly provides an incentive to further investigate the issue.

Here, we present a complete framework for scale-space analysis of 3D curves and apply it to neuronal arbors. This method, which we name *nAdder* for Neurite Analysis through Dimensional Decomposition in Elementary Regions, allows us to compute the local 3D scale along a curve, which is the size in micrometers of its 3D structure, i.e. the size at which the curve locally requires the three dimensions of space to be described. This scale is quite small for a very straight trace, and larger when the trace displays a complex and convoluted pattern over longer distances. We then propose examples and applications on several published neuronal trace datasets that demonstrate the interest of this metric to describe arbors, compare the arbors of individual neurons and extract morphological features reflecting local changes in neurite behavior. Finally we use it for a more thorough analysis of the local morphology of single neurites across the region of the whole brain of a zebrafish larvae. Implementation of the nAdder algorithms along with an array of geometry routines and functions are made available in the recently published GeNePy3D Python library [36], available at https://genepy3d.gitlab. io. Code to reproduce all figures in this study, exemplifying its use, is available at https://gitlab. com/msphan/multiscale-intrinsic-dim.

## 2 Results

### 2.1 Computation of the local 3D scale of neuronal traces from their multiscale intrinsic dimensionality

Given a neuronal arbor, we decompose each of its branches into a sequence of local curved fragments, each of a constant *intrinsic dimensionality*, defined by combining computations of both curvature and torsion of the curves. Briefly, a portion of curve with low torsion and curvature would be considered a 1D line, one with high curvature but low torsion would be approximately embedded within a 2D plane, and a curve with high torsion could only be

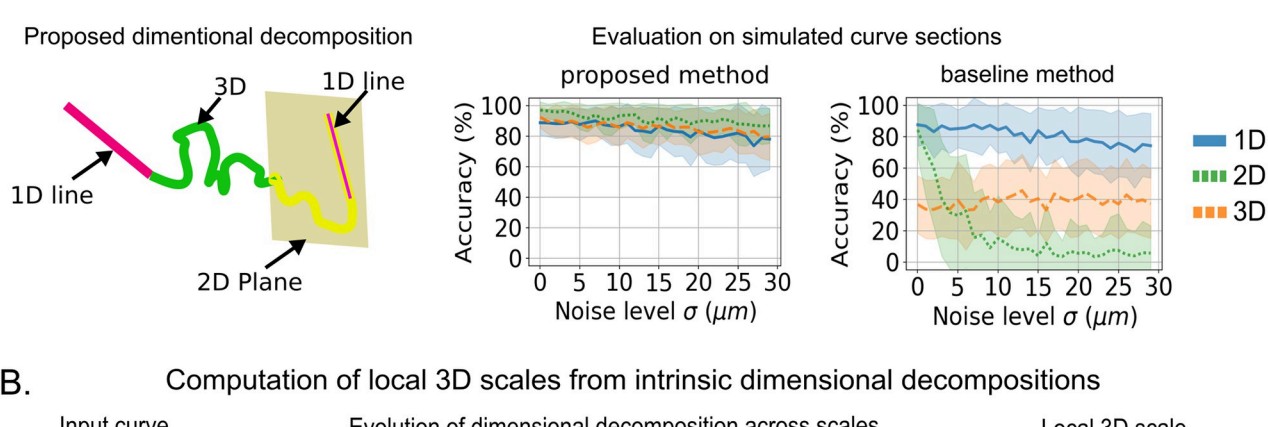

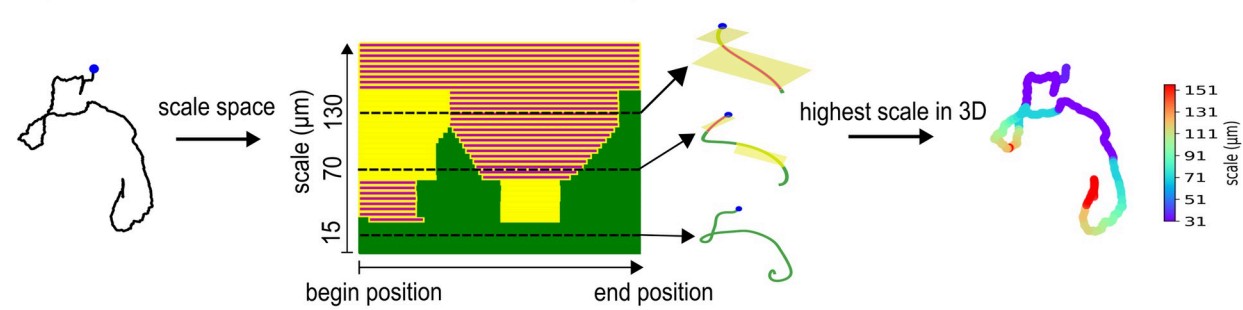

**Fig 1. Computation of the 3D local scale of neuronal traces from their multiscale dimensionality decomposition.** (A) (left) Variation of the intrinsic dimensionality along a portion of 3D curve; (right) evaluation of the decomposition on simulated trajectories comparing to a baseline method [37, 38]. Shown is the accuracy of the decomposition of each method for each dimension as additional noise is added on the simulated curve. (B) Schematic presentation of the processing of 3D curves by the nAdder algorithm.

described by taking into account the three spatial dimensions. This results in a hierarchical decomposition, capturing the fact that a 1D line is basically embedded in a 2D plane (Fig 1A and Fig A in S1 Text).

Such decomposition of traces in intrinsic dimension depends on the "scale" at which the analysis is performed. The notion of scale can be abstractly interpreted as the level of detail that an observer takes into account when considering an object, varying from a high level (when observing a trace up close i.e. at a small scale), to low level (when observing it from afar i.e. at larger scales). A scale-space is the computation of all versions of a given curve across spatial scales. Several mathematical and computational frameworks have been proposed to compute such ensemble of curves; the simplest one used here consists of smoothing the studied trace by convolutions (local averaging) of its coordinates with Gaussian kernels of increasing size. The dimensionality of such a curve element, as we analyze it at increasing scales, is typically best described as 3D at the smallest scales (i.e. taking in account a high level of detail), and becomes 2D or 1D at larger scales (i.e. low level of detail) as more and more details are smoothed out (Fig 1B, middle panel). Here, in the context of neuronal arbors, scales are measured in micrometers, by computing the radius of curvature of the smallest detail kept at that scale (see Methods for details). We first use a scale-space to compute a more robust intrinsic dimension decomposition *at a given scale*, by looking in a small interval of scale *around that scale* for the most stable decomposition (see Methods for details).

To validate the proposed decomposition, we applied it on simulated traces presenting different noise levels. Our algorithm reached accuracies above 90% with respect to the known dimensionality of the simulations at low noise level and still above 80% at high noise level, while an approach not based on space-scale which we took as baseline [37, 38] was strongly affected by noise (Fig 1A, and Fig B and C in S1 Text; details on the simulation of traces, noise levels and algorithm are available in the Methods section).

To make use of the local dimensionality decomposition *across multiple scales* and compute a simpler and more intuitive metric, we define the *local 3D scale* at a given position as the highest scale at which the trace still remains locally 3D around that position. The trace will then locally transform to 2D or 1D for scales higher than that local 3D scale. An example of local 3D scale calculation is shown in Fig 1B. This computation is done at the level of curves; to apply it to whole neuronal arbors, we decompose these arbors into curves by considering, for each 'leaf' (terminals), the curves that link it to the root (i.e. the cell body), averaging values when needed (Fig F in S1 Text). The result is an algorithm that computes a local 3D scale metric for full neuronal arbors, which we term *nAdder* (see Methods for more details on the algorithm).

Having defined a new metric and evaluated it on simulated traces, we subsequently used our algorithm to reanalyze published datasets in order to explore its capacity to extract biologically meaningful anatomical insights.

## 2.2 Application of *nAdder* to characterize neuronal arbors

To test our approach, we first examined the local 3D scales of different types of neurons presenting dissimilar arbor shapes, using traces from the NeuroMorpho.Org database (Fig 2A). We studied a mouse striatal D2-type medium spiny neuron reconstructed by [39] (Fig 2A1 and S1 Movie), and the reconstruction of a mouse retinal ganglion cell [40] (Fig 2A2 and S2 Movie), and a mouse cerebellar Purkinje neuron from [41] (Fig 2A3 and S3 Movie).

The local 3D scale computed with nAdder showed a range of values coherent with local neurite behavior, like a lower local 3D scale in straight vs. curved neurites (compare for instance region (i) and (ii) of the spiny neuron in Fig 2A1, with respective values of ∼40 μm and ∼100 μm), or a small local 3D scales in different axon portions (∼35 μm in region (iii) of the retinal ganglion neuron axon in Fig 2A2, as it runs directly towards the optic nerve head to leave the retina, compared to ∼160 μm in the proximal region (iv)). Interestingly, while most of the Purkinje neuron dendritic arbor in Fig 2A3 was characterized by low local 3D scales (∼45 μm on average), in accordance with the stereotyped planar orientation of these neuron's dendrites, and hence transformed quickly to a 2D plane when scanning the scale space, one region (v) presented local 3D scales higher than the mean value (∼80 μm), corresponding to the fact that dendrites in this region protruded in an unusual manner out of the main plane of the arbor (see S3 Movie)

We also compared our local 3D scale metric with two classic local descriptors of 3D traces, curvature and torsion (Fig G in S1 Text). Mapping of the three parameters in the Purkinje neuron presented in Fig 2A3) showed that the local 3D scale provided the most informative measure of the geometric complexity of neuronal arbors, since curvature and torsion yielded representations that were difficult to interpret as they were highly discontinuous because of local variations of the curves.

To evaluate the relevance and usefulness of the local 3D scale measure to answer more biologically meaningful questions, we then applied *nAdder* to compare neuronal arbors at different stages of their development and in normal vs. experimentally altered contexts. We selected data from [26] hosted in the NeuroMorpho.Org database which describe the expansion of the

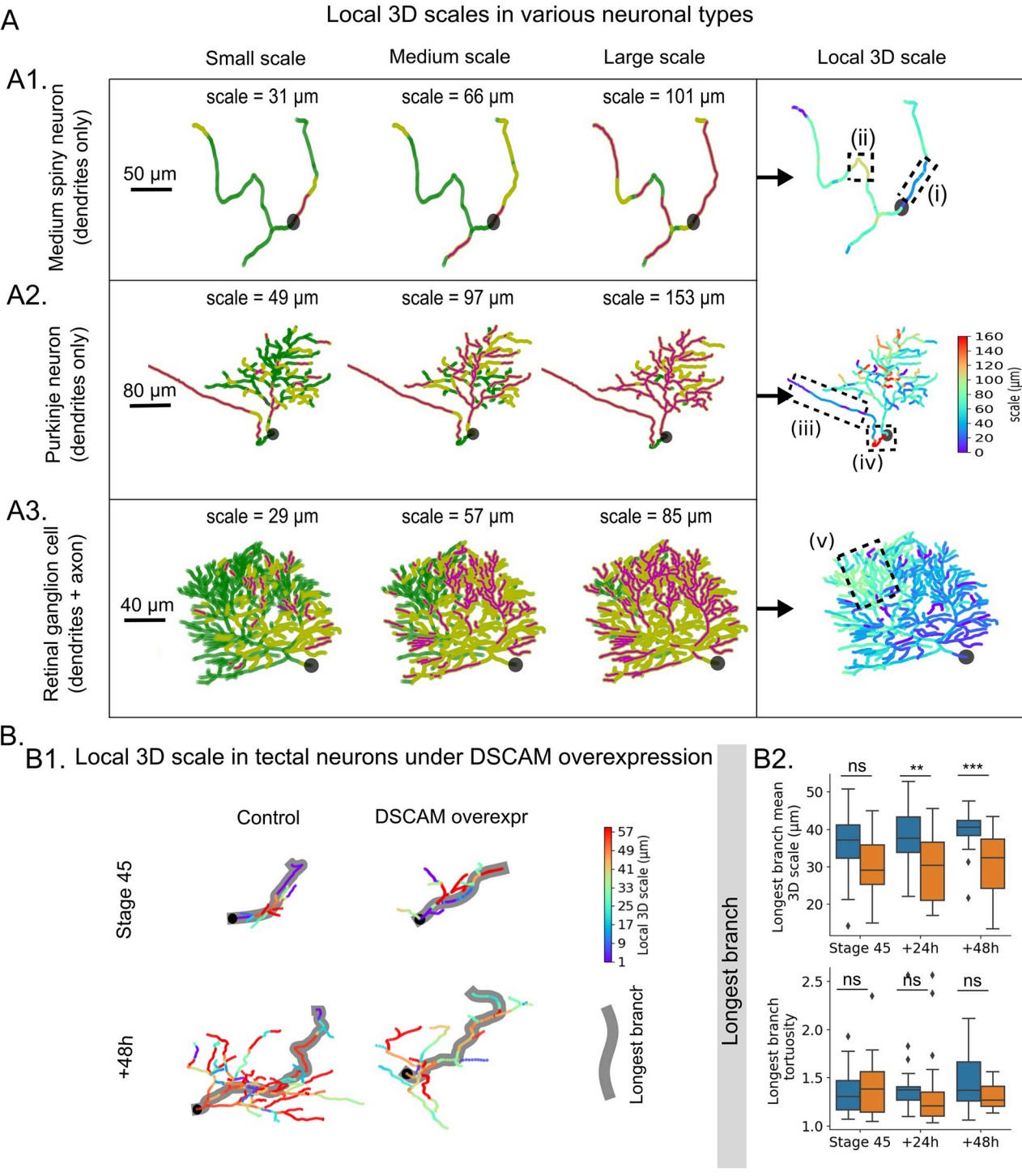

**Fig 2. Local 3D scale analysis of various neuronal traces.** (A) Application of nAdder to three types of mouse neurons with different arbor shape and size: (A1) striatal D2 medium spiny neuron [39], (A2) retinal ganglion cell [40], (A3) cerebellar Purkinje neuron [41]. The first three columns show the intrinsic dimensionality decompositions of the neurons at small, medium and large scales. The local 3D scale, computed from suites of such decomposition across multiple scales, is shown in the last column. The maximum scale is set at 160 μm based on the longest branch analyzed. Dotted line boxes (i-v) frame areas of interest discussed in the text. (B) Dendritic arbor traces from X. laevis tectal neurons were obtained from [26]. (B1) Examples of local 3D scale maps from control (left) and DSCAM-overexpressing (DSCAM overexpr) neurons at Stage 45, and 48 hr after initial imaging. The maximum scale is set to 60 μm based on the mean length of the arbor's longest branch (highlighted in gray). The cell body's position is indicated by a black square. (B2) Mean local 3D scale (above) and tortuosity (bellow) of the longest branch. Two-way ANOVA, Student's t-test with Holm-Sidak for multiple comparisons were used, * p≤0.05, ** p≤0.01, *** p≤0.005, **** p≤0.001.

dendritic arbors of the Xenopus laevis tectal neuron during development and the effect of altering the expression of Down syndrome cell adhesion molecule (DSCAM). In this study, the authors showed that downregulation of DSCAM in tectal neuron dendritic arbors increases the total dendritic length and number branches, while overexpression of DSCAM lowers them. We reanalyzed neuronal traces generated from the dendritic arbors in this study by computing their local 3D scale using *nAdder*.

Our analysis showed that the mean local 3D scale of developing X. laevis tectal neurons overexpressing DSCAM is smaller than that of control neurons, with a significant difference between the two conditions, 48 hr after the start of the observations at Stage 45 (Fig 2B1 and 2B2, above); this result is consistent with the original analysis by [26], indicating that DSCAM overexpression leads to more simple arbor morphologies, as measured by the number of branches and length of reconstructed arbors. In their paper, the authors also studied tortuosity, the ratio of curvilinear to Euclidean length of a trace, a global morphological metric classically used to measure geometric complexity. While the authors show that DSCAM downregulation increases the tortuosity of the neuron's longest dendritic branch (see Fig 4 of [26]), they did not report this parameter for DSCAM overexpression. We computed the tortuosity of the longest branch and indeed observed no significant difference between control and DSCAM overexpressing neurons at any stage of the analysis (Fig 2B2, below). On the contrary, computation of the local 3D scale show a significant difference (Fig 2B2, above). Overall, the local 3D scale metric computed with *nAdder* offers a new local measure of the geometric complexity of neuronal traces, and provides information robust across neuron types, shapes and sizes. Our analysis shows that measuring the neurites' local 3D scale provides biologically relevant information and can measure subtle differences in the complexity of their trajectory that are not apparent with classic metrics.

## 2.3 Local 3D scale of individual neurites across the whole larval zebrafish brain

We next sought to test *nAdder* over a large-scale 3D dataset encompassing entire long-range axonal projections. A database including 1939 individual neurons traced across larval zebrafish brains and co-registered within a shared framework has been published by [12]. Here, we reanalyzed this dataset by computing the local 3D scales of all available neuronal traces, focusing on projections linking distinct brain regions (i.e. that terminated in a region distinct from that of the cell body), interpreted as axons. We then explored the resulting whole-brain map of the geometric complexity of projection neurons.

**Local 3D scale variations across brain regions.**   Processing of the 1939 traces of the larval zebrafish atlas with *nAdder* resulted in a dataset where the local 3D scale of each inter-region projection (i.e. projecting across at least two distinct regions), when superimposed in the same referential, could be visualized (Fig 3A and Fig H panel A in S1 Text). Coordinated variations in local 3D scale values among neighboring traces were readily apparent on this map, such as in the retina (see region (i) in Fig 3A) and torus semicircularis (ii) which presented high local 3D scale values, or the octaval ganglion (iii) characterized by low values.

To quantify these local differences, we computed the mean local 3D scale of inter region projections for each of the 36 brain regions defined by Kunst et al. This enabled us to establish an atlas of the local 3D scale of fish axons highlighting variations of their geometric complexity in different brain regions (Fig 3B and Fig H panel B in S1 Text). Similar atlases could be derived for all projection subsets originating from (Fig I panel A in S1 Text), passing through (Fig J panel A in S1 Text) or terminating in a region (Fig K panel B in S1 Text), again showing inter-regional variations. The distributions of local 3D scale within each region nevertheless

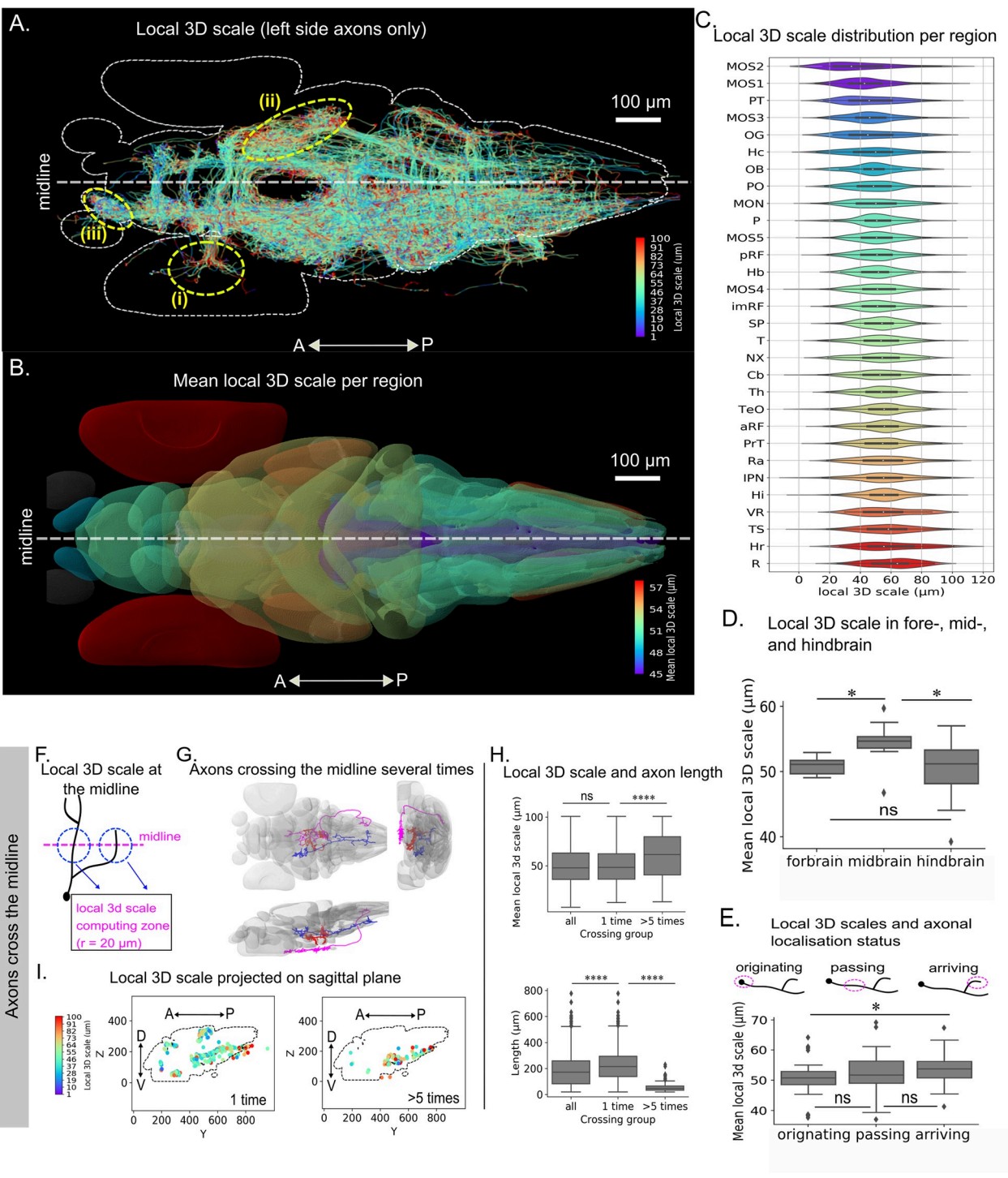

**Fig 3. Local 3D scale mapping across the whole larval zebrafish brain.** Traces analyzed correspond to those presented in [12]. (A) Local 3D scale analysis of all axonal traces originating from neurons in the left hemisphere; the maximum scale is set to 100 μm based on the width of the largest brain region. (B) Mean local 3D scale by brain regions. Regions with fewer than 15 traces were excluded (in gray). Values were clipped from the 5th to 95th percentiles for clearer display. (C) Distribution of the local 3D scale values in each brain region (see Table A of S1 Text for abbreviation definitions). (D) Mean local 3D scale in fore-, mid- and hindbrain regions (E) Mean local 3D scale of axons originating from, passing through, and arriving in each brain region. (F) Example of axonal arbors with many branches crossing the midline. (G) Diagram summarizing computation of the local 3D scale at the midline. (H) Local 3D scale (top) and length (bottom) of all axons crossing the midline vs. those crossing only once and more than five times. (I) View of the midline sagittal plane showing the local 3D scale of axons crossing only once and more than five times. The maximum scale is set to 100 μm based on the width of the largest brain region analyzed. Wilcoxon tests with Holm-Sidak correction for multiple comparison was used, * p≤0.05, ** p≤0.01, *** p≤0.005, **** p≤0.001. Color scale in C corresponds to that used in B.

showed a high variance (Fig 3C, and panel B of Figs I,J and K of S1 Text), reflecting the diversity of neuronal types and variations in the local structure of their axons as they exit or enter a brain region. This variability is illustrated in Fig L of S1 Text, showing three regions, the Medulla Oblongata Strip 3 (MOS3), Subpallium (SP) and Torus semicircularis (TS), respectively presenting different patterns exhibited by axons originating from, passing through and arriving in these regions.

It was observed in [12] that the connectivity strength in the mid- and hindbrain are stronger than that in forebrain. The distributions of local 3D scale computed in the fore-, mid- and hindbrain regions show a higher average value in mid- compared to fore- and hindbrain regions (Fig 3D), although the difference in not significant due to the small number of regions. The difference is clearer (while still not statistically significant) for axons terminating in a region (Fig K panel C of S1 Text), and all but absent for axons with soma in these regions and passing axons (Figs I and J panel C of S1 Text).

Comparing the local 3D scale of the three types of axons showed that inter-regional axons terminating in a region had on average higher local 3D scales than those originating from or passing through a region (Fig 3E). This observation is consistent with the idea that the proximal axonal path and terminal arbor differ in their complexity, as related to their developmental assembly and function.

When comparing local 3D scale with number of branching points and axonal length across brain regions, we found potentially significant correlations for axons having soma in the region (Fig I panel D-E of S1 Text), but not for passing axons (Fig J panel D-E of S1 Text) and axons terminating in the region (Fig K panel D-E of S1 Text). This suggests that highly branched and long axons could exhibit more complex shapes than short and no-branched axons.

Another class of axons we chose to focus on are commissural axons, that cross the midline. They play a major role by interconnecting the two hemispheres and have been well studied, in particular for the specific way by which they are guided across the midline [42]. Examining axonal arbors intersecting the midline in the whole larval fish dataset, we found that a majority (71%) crossed only once, making a one-way link between hemispheres, while a significant proportion crossed multiple times, with 10% traversing the midline more than 5 and up to 24 times (Fig 3F and Fig M of S1 Text). Most individual branches (i.e. neuronal arbor segments between two branching points, or between a branching point and the cell body/terminal) crossed the midline just once, meaning that neurons crossing multiple times did not do so with long meandering axons but with many distinct branches of their arbor (Fig M panel B of S1 Text). Examples of such axonal arbors crossing the midline more than 10 times are shown in Fig 3F. Measuring the local 3D scale of crossing axon branches in a 20 μm range centered on the midline (Fig 3G), we observed higher values for the ones traversing the midline several times compared to those only crossing once (Fig 3H, top). Axon branches crossing the midline multiple times were also much shorter than those crossing only once (Fig 3H, bottom). They localized mostly in the hindbrain in a ventral position, while single crossings were frequent in the forebrain commissures (Fig 3I). This is consistent with the existence of two populations of commissural axons in the vertebrate brain, the hindbrain being home to neurons following complex trajectories across the midline with many short and convoluted branches, and forebrain commissural neurons forming longer and straighter axons crossing fewer times and often only once. Given the lack of clear link between the length and local 3D scale of crossing branches (Fig M panel C of S1 Text) the full picture is most likely more complex and would need more complete study including functional data to be clarified.

Together, these results show that the local 3D scale computed with *nAdder* is informative at the whole brain level, adding a new level of description for analyzing connectomic datasets.

**Characterization of axon behavior at defined anatomical locations and along specific tracts.**    A key feature of our metric is its local nature, i.e. its ability to inform on the geometry of single neurites not only globally, but at each point of their trajectory. To illustrate this, we used *nAdder* to analyze the trajectories of Mitral cell axons originating in the olfactory bulb (OB) and projecting to multiple destinations in the pallium (P), subpallium (SP) or habenula (Hb) [43]. Here again we found variations in axonal complexity across regions (Fig 4A), with relatively low local 3D scales in the OB (∼40 μm) and Hb formation (∼37 μm) compared to higher values in the P (∼50 μm) and SP (∼47 μm).

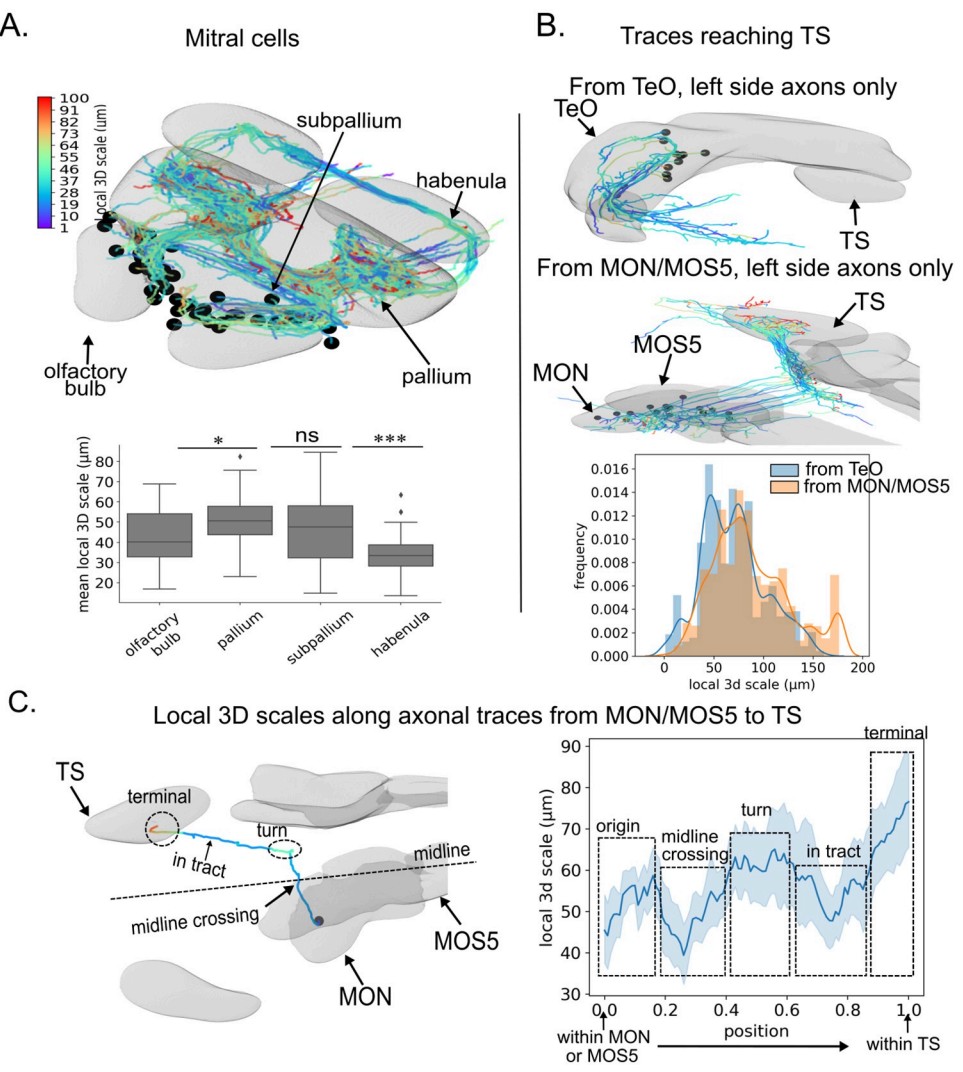

**Fig 4. Local 3D scale along different axonal populations.** (A) Local 3D scale variation along axons of mitral cells. (B) *nAdder* analysis of TS-terminating axons originating from the Optic Tectum (TeO) (top) and Medial Octavolateral Nucleus (MON) or Medulla Oblongata Strip 5 (MOS5) (middle). Comparison of the distributions of local 3D scale of axons coming from the two regions (bottom). (C) Local 3D scale variations along axons coming from MON/MOS5. Left, example view of one axon indicating different segments of interest. Right, quantification (n = 32 axons). The maximum scale in A is set to 100 μm based on the width of the largest brain region analyzed. The maximum scale in B is set to 175 μm based on the mean length of the longest branch. Only the longest branches are included in the computation. Wilcoxon tests with Holm-Sidak corrections for multiple comparison were used, * p≤0.05, ** p≤0.01, *** p≤0.005, **** p≤0.001.

Finally, we investigated local 3D scale across different axonal populations innervating a given brain region. We performed this analysis in the Torus Semicircularis (TS) which presents a high average local 3D scale, with many complex axonal traces (Fig L of S1 Text, last row). Most of these traces corresponded to axons with terminations in the TS (70% vs. 5% axons emanating from TS and 25% axons passing through TS). Focusing on this population, we selected three groups of axons originating from the Tectum (TeO), Medial Octavolateral Nucleus (MON) and Medulla Oblongata Stripe 5 (MOS5), respectively, excluding other regions from which only a small number ($< 10$) of incoming axons had been traced (a detailed list of all regions with traces terminating in the TS is in Table B of S1 Text). Examining the distribution of local 3D scale inside the TS, we observed that tectal axons overall had a simpler structure, characterized by lower local 3D scale values than those coming from the MON/MOS5 (Fig 4B). In particular, complex axons with a local 3D scale above 150 µm mostly originated from the MON/MOS5 (Fig 4B, below). This difference, revealed by *nAdder*, points to the existence of divergent developmental mechanisms of pathfinding and/or synaptogenesis for these two populations of axons within a shared target area.

We then focused on axons projecting from the MON/MOS5 to the TS. We computed the local 3D scale of these axons at all coordinates of their trajectory from the MON/MOS5 to the TS, normalized with respect to total length (Fig 4C). We found that these axons' local 3D scale was highly correlated and that its average value varied widely along their course from the MON/MOS5 to the TS: low at the start, it increased inside the MON/MOS5 before dropping when the axons crossed the midline. It then increased again as they made a sharp anterior turn, dropped again within the straight tract heading to the TS, before rising to maximum values inside the TS. This result provides a striking example of stereotyped geometrical behavior among individual axons linking distant brain areas, likely originating from a same type of neurons, that our metric is able to pick up and report.

Overall, our analysis of the larval zebrafish brain atlas using *nAdder* highlight both the widespread heterogeneity of the local geometrical complexity of axons and its correlation with brain areas, neuron type or position within a trace.

## 3 Materials and methods

### 3.1 Intrinsic dimension decomposition of neurite branches based on scale-space theory

**Intrinsic dimension decomposition.** Let's consider a 3D parametric curve $\gamma(u) = (x(u), y(u), z(u))$ for $u \in [0, 1]$; the *intrinsic dimensionality* of that curve can be defined as the smallest dimension in which it can be expressed without significant loss of information. An *intrinsic dimension decomposition* is a decomposition of a 3D curve into consecutive fragments of different intrinsic dimensionality, i.e. lying intrinsically on a 1D line, 2D plane or in 3D. Note that such decomposition needs not be unique. For example, not taking scales into account, two consecutive lines followed by one plane can in theory be decomposed into one 3D fragment, one 1D and one 3D fragments, two consecutive 1D and one 2D fragments, or two consecutive 2D fragments (Fig A of S1 Text). The last two decomposition schemes are both meaningful and their combination represents a hierarchical decomposition of $\gamma$ where linear fragments are parts of a larger planar fragment.

Here, we compute the decomposition of a curve intrinsically by looking at the curvature and torsion. Let us define the curvature $\kappa$ of $\gamma$ by $\kappa = \|\gamma' \times \gamma''\| / \|\gamma'\|^3$, corresponding to the inverse of the radius of the best approximation of the curve by a circle locally, the osculating circle. The torsion $\tau$ of $\gamma$ is $\tau = ((\gamma' \times \gamma'') \cdot \gamma''') / \|\gamma' \times \gamma''\|^2$ and corresponds to the rate of change of the plane that includes the osculating circle. We then determine that if $\kappa$ is identically equal

to zero on a fragment then that fragment is a 1D line; similarly $\tau$ being identically equal to zero defines a 2D arc curve. Thus we define a linear indicator $L$ of $\gamma$:

$$L(u) = \begin{cases} 1, & \text{if } \kappa(u) \leq \varepsilon_\kappa \\ 0, & \text{otherwise,} \end{cases}$$

and a planar indicator $H$ of $\gamma$:

$$H(u) = \begin{cases} 1, & \text{if } |\tau(u)| \leq \varepsilon_\tau \\ 0, & \text{otherwise,} \end{cases}$$

where $\varepsilon_\kappa$ and $\varepsilon_\tau$ are the tolerances for computed numerical errors. We note that only using the $H$ indicator is not sufficient to estimate the 2D plane. For example, a 2D plane fragment composed of a 1D line followed by a 2D arc cannot be entirely identified as 2D since the torsion is not defined as the curvature tends to 0. We therefore consider the planar-linear indicator $T = L \cup H$ instead of $H$ for characterizing the curve according to such hierarchical order.

**Scale space.**   In practice, the resulting dimension decomposition is closely tied to a 'scale' at which the curve is studied, i.e. up-close all differentiable curves are well approximated by their tangent and thus would be linear. That property have been formalised through *Scale-spaces*, which have been described for curves in 2D [44] or 3D [35] in particular, to give a robust meaning to that intuition. A scale-space is typically defines as a set of curves $\gamma_s$ such that $\gamma_0 = \gamma$ and increasing $s$ leads to increasingly simplified curves. The scaled curve can be calculated by convolving $\gamma$ with a Gaussian kernel of standard deviation $s$ [45], or by using mean curvature flow [46]. In 2D, it has been shown for example that a closed curve under a mean curvature motion scale space will roundup with increasing s and eventually disappear into a point [47]. In the following we use a gaussian scale-space on each coordinate, i.e. $\gamma_s$ is computed by convolving $\gamma$ by a 1D gaussian function of width $s$, $\mathcal{N}(0, s)$.

We first use the scale space to compute the defined decomposition "around" a scale of interest to have a more stable solution. Let $S$ be a given set of scales around a scale of interest $s_\mu$ (for example $S = [s_\mu - \delta s, s_\mu + \delta s]$), we compute first, for all $s \in S$, $L_s$ and $T_s$ by calculating the curvatures $\kappa_s$ and the torsions $\tau_s$. We then exclude fragments (pieces of the curve with a given, constant dimentionality) smaller than a length threshold $\varepsilon_\omega$ to eliminate small irrelevant fragments. From $L_s$ and $T_s$, the linear fragments denoted as $\mathcal{D}_{L_s}$ and planar-linear fragments denoted as $\mathcal{D}_{T_s}$ are deduced across all $s \in S$.

In a second step, we estimate the most durable combination of fragments from the linear segments $\mathcal{D}_{L_s}$ and linear-planar segments $\mathcal{D}_{T_s}$ calculated in the first step. We compute the number of fragments at every $s \in S$, to measure how long each combination of fragments exists. We then select the combination of fragments remaining for the longest subinterval of $S$. Knowing that each fragment in the combination can have different lengths among that subinterval, we thus select the longest candidate. In cases where fragments overlap, we split the overlap in half. Of note, we repeat this step twice, first to estimate the best combination of linear-planar segments from $\mathcal{D}_{T_s}$ input and mark the corresponding subinterval, then to estimate the best combination of linear fragments from $\mathcal{D}_{L_s}$ input within that subinterval. The final result is the hierarchical dimension decomposition of the curve $\gamma$ around a given scale of interest $s_\mu$. Fig D of S1 Text show an illustration of the decomposition of a curve across $s$ in consecutive planar/nonplanar fragments, then the linear fragments are identified within each planar fragment.

However, the definition of $S$ via a scale of interest and its neighborhood is not straightforward. A scale defined as the standard deviation of the Gaussian kernel as above for example will be influenced by the kernel length and the curve sampling rate, and would be unintuitive to set, and fixed values of $\delta s$ would be arbitrary. To make the scales become a more relevant physical/biological measurement, we employ as scale parameter the radius of curvature $r_\kappa$, defined as the inverse of the curvature $\kappa$, in $\mu$m. For a given curve at a given scale, the level of detail kept at that scale can be characterized by the maximal $r_\kappa$ along that curve; it would correspond to the smallest bump, or tighter turn, of the curve. Scales keeping small $r_\kappa$ represent high level of detail looking at small objects such as small bumps and tortuosity, while scales smoothing the curve out and keeping only larger $r_\kappa$ represent lower level of detail, keeping only larger object and features such as plateaus or turns.

Thus, to ease the interpretation and usage of scale spaces, we index them in micrometer by associating to a scale in micron $\hat{r}_\kappa$ a set $S$ of standard deviation of the gaussian scale space. We determine, for a given curve, for each point $u$ whose $r_\kappa(u) < \hat{r}_\kappa$ the standard deviation $s$ such that $r_{\kappa_s}(u) = \hat{r}_\kappa$. This therefore associates, for each curve, the anatomically relevant scale of interest in micron $\hat{r}_\kappa$ with the list $S$ of standard deviations that lead to the points of that curve to have a curvature radius smaller than $\hat{r}_\kappa$. We can now define the scale $\hat{r}_\kappa$ in $\mu$m and use $S$ to estimate the dimension decomposition of $\gamma$ as define above.

## 3.2 Evaluation of dimension decomposition on simulated curve

The simulated curve consists of consecutive 1D lines, 2D planes and 3D regions. The simulation of 1D lines is done by simply sampling a sequence of points on an arbitrary axis (e.g. $x$ axis), then rotating it in a random orientation in 3D. The simulation of random yet regular 3D fragments embedded in a 2D plane is more challenging. We tackle this issue by using the active Brownian motion model [48]. Active Brownian motion of particles is an extended version of the standard Brownian motion [49] by adding two coefficients the translational speed to control directed motion and the rotational speed to control the orientation of the particles. At each time point, we generate the new coordinates $(x, y, 1)$ by the following formulas:

$$D_T = \frac{k_B T}{6 \pi \eta R}$$

$$D_R = \frac{k_B T}{8 \pi \eta R^3}$$

$$\frac{d}{dt} \varphi(t) = \Omega + \sqrt{2 D_R}\, W_\varphi$$

$$\frac{d}{dt} x(t) = v \cos \varphi(t) + \sqrt{2 D_T}\, W_x$$

$$\frac{d}{dt} y(t) = v \sin \varphi(t) + \sqrt{2 D_T}\, W_y,$$

where $D_T$ and $D_R$ are the translational and rotational coefficients, $k_B$ the Boltzmann constant, $T$ the temperature, $\eta$ the fluid viscosity, $R$ particle radius, $\varphi$ rotation angle, $\Omega$ angular velocity and $W_\varphi$, $W_x$, $W_y$ independent white noise. After generating the simulated intrinsic 2D fragment, a random rotation is applied. For simulating a 3D fragment, we extended the Active

Brownian motion model in 2D [48] to 3D as follows:

$$\frac{d}{dt}\,\varphi(t) = \cos\Omega + \sqrt{2\,D_R}\,W_\varphi$$

$$\frac{d}{dt}\,\theta(t) = \sin\Omega + \sqrt{2\,D_R}\,W_\theta$$

$$\frac{d}{dt}\,x(t) = v\cos\theta(t)\sin\varphi(t) + \sqrt{2\,D_T}\,W_x$$

$$\frac{d}{dt}\,y(t) = v\sin\theta(t)\sin\varphi(t) + \sqrt{2\,D_T}\,W_y$$

$$\frac{d}{dt}\,z(t) = v\cos\varphi(t) + \sqrt{2\,D_T}\,W_z,$$

where $(\varphi, \theta)$ are spherical angles. We simulate the curve $\gamma$ with a sequence of $n_\omega$ consecutive fragments of varying random intrinsic dimensions. The curve $\gamma$ is then resampled equally with $n_\gamma$ points and white noise $\mathcal{N}(0, \sigma^2)$ is added. We set $n_\omega \leq 5$, $n_\gamma = 1000$ points and vary $\sigma$ between 1 and 30 $\mu m$. The upper value of $\sigma = 30\mu m$ is high enough to corrupt local details of a simulated fragment with about 100 $\mu m$ of length in our experiment.

The metric used to measure the accuracy of the intrinsic dimension decomposition, defined to be between 0.0 and 1.0 and measures an average accuracy across fragments, is given by $(1/n_\omega)\sum_i \max_j \xi(\omega_i, \hat{\omega}_j)$, where $\{\omega_1, \omega_2, \ldots, \omega_{n_\omega}\}$ are the simulated intrinsic fragments, $\{\hat{\omega}_1, \hat{\omega}_2, \ldots, \hat{\omega}_{n\hat{\omega}}\}$ the estimated intrinsic fragments and $\xi(., .)$ the $F_1$ score [50] calculated by:

$$2 \times \frac{Precision \times Recall}{Precision + Recall},$$

where

$$Precision = \frac{|\omega \cup \hat{\omega}|}{|\omega|} \qquad \text{and} \qquad Recall = \frac{|\omega \cup \hat{\omega}|}{|\hat{\omega}|}.$$

To evaluate the intrinsic dimensions decomposition algorithm independently of the nAdder algorithm we have to choose a scale at which to compute the decomposition. We either take the one corresponding to the largest accuracy (optimal scale, Fig B of S1 Text) or one at a fixed scale (Fig C of S1 Text), $r_k = 20$ $\mu m$, small enough to avoid deforming the simulated curve. Our method is then compared with a baseline method [37, 38] that first iteratively assigns each point on the curve as linear/nonlinear by a collinearity criterion, then characterize nonlinear points as planar/nonplanar by a coplanarity criterion. For both methods, the curve was first denoised based on [51].

## 3.3 Local 3D scale computation

Assuming *a set* of scale of interest, the local 3D scale of a point along a curve is the smallest scale at which the local scale is not 3D anymore, computed by checking at each scale the intrinsic dimension of that point. While the dimension of most points decrease with scales, it does not have to be monotonous and in some cases the dimension of a point revert back to 3D from being 2D/1D, as we increase scales. We therefore select the first scale of the longest 2D/1D subinterval as local 3D scale. In case no subintervals are found, it means that the point remains 3D across the whole interval and its local 3D scale is set as the highest scale.

An example of this sequence of dimension decompositions is illustrated in Fig D of S1 Text where $\mathcal{D}_{L_s}$ and $\mathcal{D}_{T_s}$ are superimposed. The curve is completely 3D at low $s$ value, several linear and planar fragments then appear at higher $s$, progressively fuse to form larger linear and planar fragments and eventually converge to one unique line at very high value of $s$. Thus An interpretation of the local 3D scale is that it is the size of the largest motif left at the scale at which the 3D structure has disappeared. In the following experiments, we took regularly spaced scale of interests from 0 to a maximum, indicated in each case. A summary descritpion of the nAdder algorithm is available in Algorithm A in S1 Text.

The nAdder algorithm is theoretically influenced by the initial sampling of the curve used in practice for numerical computation. Fig E of S1 Text shows that while specific values are affected by changes in sampling rate, the overall behavior as well as local differences are robust to those changes. In all the following experiments we took a constant sampling equal to $1\mu$m.

### 3.4 Local 3D scale analysis of neuronal traces

We consider a neuronal trace as a 3D tree and first resample it with a spacing distance between two points equal to 1 $\mu$m. B-spline interpolation [52] of order 2 is used for the resampling. We then decompose the neuronal arbor into potentially overlapping curves, since the intrinsic dimension decomposition and local 3D scale are computed on individual neuronal branches. Several algorithms are available for tree decomposition. We explored three, functioning either by longest branch (first taking the longest branch, then repeating this process for all subtrees extracted along the longest branch), by node (taking each curve between two branching nodes) or by leaf (taking, for each leaf, the curve joining the leaf to the root). Local scales were computed for all three considered decompositions and the 'leaf' decomposition was found to give the most robust and meaningful results as shown in Fig F of S1 Text. That method gives partially overlapping curves; thus, we compute the average of the values when needed; we verified that the standard deviations were small.

Next, the local 3D scale is computed for each extracted sub-branch within a scale interval of interest. The scale is defined as the radius of curvature and can be selected based on the size of the branches or that of the region studied. For example, in the case of Xeonopus Laevis tadpole neurons (Fig 2B), we selected scales from 1 to 60 $\mu$m which was sufficient to study neuronal branches of about 180 $\mu$m in length (i.e. a 60 $\mu$m radius of curvature corresponds to a semicircle with length equal to $60\pi \approx 180$ $\mu$m). In the case of the zebrafish brain dataset (Fig 3), we selected scales from 1 to 100 $\mu$m based on the length of the largest region, which was about 300 $\mu$m. When studying axons arriving to the Torus Semicircularis (Fig 4), we set the maximum scale to 175 $\mu$m as the neuronal branches are on average 525 $\mu$m long. Moreover, we excluded the sub-branches whose lengths are smaller than a threshold set to 5 $\mu$m, which is half the size of the smallest brain region. Branches shorter than 5 $\mu$m contribute very little to the local 3D scale result.

### 3.5 Toolset for neuronal traces manipulation

We provide an open source Python toolset for seamless handling of 3D neuronal traces data. It supports reading from standard formats (swc, csv, etc), visualizing, resampling, denoising, extracting sub-branches, calculating basic features (branches, lengths, orientations, curvatures, torsions, Strahler order [53], etc), computing the proposed intrinsic dimension decompositions and the local 3D scales. The toolset is part of the open source Python library GeNePy3D [36] available at https://genepy3d.gitlab.io/ which gives full access to manipulation and interaction of various kinds of geometrical 3D objects (trees, curves, point cloud, surfaces).

**Table 1. Origin of the traces reanalysed in this study.**

| Name | Source article | Donwload page | Search key | Note |
|---|---|---|---|---|
| Medium spiny neuron | [39] | http://neuromorpho.org | by neuron name: *3817_CPi_PHAL_Z001_app2_split_34* | Section 2.1 |
| Purkinje neuron | [41] | http://neuromorpho.org | by neuron name: *Purkinje-slice-age P43–6* | Section 2.1 |
| Retina ganglion cell | [40] | http://neuromorpho.org | by neuron name: *Badea2011Fig2Ca-R* | Section 2.1 |
| Xenopus laevis | [26] | http://neuromorpho.org | by archive: *Cohen-Cory* | Section 2.2 |
| Zebrafish atlas | [12] | https://fishatlas.neuro.mpg.de/neurons/download | by neuron groups: *Kunst et al* | Section 2.3 |

### 3.6 Analyzed neuronal traces datasets

The traces used in our studies are all open and available online for downloading. They consist of neuronal reconstructions from NeuroMorpho.Org and the Max Planck Zebrafish Brain Atlas listed in the Table 1.

## 4 Discussion

We presented a novel method for the multiscale estimation of intrinsic dimensions along an open 3D curve and used it to compute a local 3D scale, which we propose as a new local metric to characterize the geometrical complexity of neuronal arbors. We demonstrated that our method, dubbed *nAdder*, is accurate and robust, and applied it to published trace data, showing its relevance and usefulness to study neurite trajectories from the level of single neurons to the whole brain.

Our new approach provides a solution to the problem of measuring the geometrical complexity of 3D curves not just globally but also locally, i.e. at successive points of their trajectory. It enables to compute the dimensionality of such curves at any position, for a given scale of analysis. By scanning the scale space, one can then determine the value at which the curve requires all three dimensions of space for its description, a sampling-independent local metric which we term the local 3D scale. Mathematically, scale spaces of curves are trickier to implement in 3D than in 2D and have been much less studied. In 2D for example, scale spaces based on curvature motion are known to have advantageous properties compared to Gaussian ones: they arise naturally from principled axiomatic approaches, are well described mathematically and can be computed through well characterized numerical solutions of partial differential equations [32]. They are readily extended to 3D surfaces by using principal curvatures [54], but no simple equivalent is known for 3D curves. One theoretical issue is that using only the curvature to determine the motion of 3D curves would not affect the torsion, and for example a set of increasingly tighter helices would have increasingly higher curvature but identical torsion. Here, we provided and thoroughly evaluated a practical solution to this problem by associating a spatial scale to an ensemble of Gaussian kernels. In future work, additional theoretical studies of 3D scale spaces of curves could potentially lead to simpler and more robust scale space algorithms with solid mathematical foundations which could advantageously replace Gaussian scale space.

The *nAdder* approach fills a gap in the methodologies available to analyze neuronal traces, which until now were lacking a straightforward way to measure their geometrical complexity and its variations, both within and between traces. It is complementary to techniques classically used to study the topology of dendritic and axonal arbors (number and position of branching points, for example), providing a geometrical dimension to the analysis that is a both simpler and more robust than the alternative direct computation of curvature and torsion

[23]. In practice, the local 3D scale of neurons (as measured with nAdder and observed in this study) ranges from 0 to 100–200 $\mu$m. The lowest values are typically found in axonal tracts that follow straight trajectories, or in special cases such as the planar dendritic arbor of cerebellar Purkinje neurons shown in Fig 2A3. Higher, intermediate values are for instance observed at inflexion points along tracts where axons change course in a concerted manner. The highest values of local 3D scale typically characterize axonal arbor portions where their branches adopt a more exploratory behavior, in particular in their most distal segments where synapses form. A striking example is given in Fig 4C, showing the trajectory of an axon from the MON/MOS5 to the torus semicircularis, where changes in local 3D scale correlate with distinct successive patterns: a straight section through the midline, a sharp turn followed by another straight section before more complex patterns in the target nucleus. Importantly, nAdder not only enables to quantify how geometric complexity varies along a single axon trace, but also to compare co-registered axons of a same type and to characterize coordinated changes in their behavior (Fig 4C, right panel). It thus offers a way to identify key points and stereotyped patterns along traces. This will be of help to study and hypothesize on the developmental processes at the origin of these patterns, such as guidance by attractive or repulsive molecules [55], mechanical cues [56], or pruning [57, 58].

The high values of local 3D scale near-systematically observed in the most distal part of axonal arbors are also of strong interest, as these segments typically undergo significant activity-based remodeling accompanied by branch elimination during postnatal development [57], resulting in complex, convoluted trajectories in adults [29, 30]. Further integrating topological and geometrical analysis and linking these two aspects with synapse position could then be a very beneficial, if challenging, extension of *nAdder*. To this aim, one would greatly benefit from methods enabling both faithful multiplexed axon tracing over long distances and mapping of the synaptic contacts that they establish. Progress in volume electron [5], optical [9] or X-ray holographic microscopy [59] will be key to achieve this in vertebrate models.

Beyond neuroscience, we also expect *nAdder* to find applications in analyzing microscopy data in other biological fields wherever 3D curves are obtained. For example, it could help characterizing the behavior of migrating cells during embryogenesis [60], or be used beside precise biophysical models to interpret traces from single-particle tracking experiments [61].

Importantly, an implementation of the proposed algorithms, along with all codes to reproduce the figures, is openly available, making it a potential immediate addition to computational neuroanatomy studies. Specifically, *nAdder* is part of GeNePy3D, a larger quantitative geometry Python package providing access to a range of methods for geometrical data management and analysis. More generally, it demonstrates the interest of geometrical mathematical theories such as spatial statistics, computational geometry or scale-space for providing some of the theoretical concepts and computational algorithms needed to transform advanced microscopy images into neurobiological understanding.

## Supporting information

**S1 Text. Supporting information. Fig A: Different schemes for the intrinsic decomposition of 3D traces.** (A) The green trace is entirely 3D. (B) The trace is decomposed by 1D line (pink) followed by 3D fragment (green). (C) The trace is decomposed by suite of 1D lines and 2D plane. (D) The trace is decomposed as sucsessive 2D planes. The decompositions are hierarchial as a 1D line is lying on a 2D plane, and the 2D plane itself is lying within a 3D portion. **Fig B: Evaluation of intrinsic dimensional decomposition on simulated 3D traces.** (A) Precision and Recall of the proposed nAdder algorithm and the baseline [37, 38] as a function of the noise level $\sigma$ varying between 1 and 30 $\mu$m. The algorithms are applied at various scales

from 1 to 100 $\mu$m and the scale with the largest accuracy (optimal scale) is chosen. (B) Comparison of estimated intrinsic dimensions at four different noise levels. Compared with the baseline approach, the nAdder is more robust to noise and gives much higher accuracies in both Precision and Recall. Details of the algorithms and simulations are shown in Methods. **Fig C: Evaluation of the dimensionality decomposition algorithm at a fixed scale.** Accuracy, Precision and Recall of the nAdder algorithm and the baseline approach from [37, 38] as a function of the noise level $\sigma$ varying between 1 and 30 $\mu$m. The algorithms are applied at a fixed scale = 20 $\mu$m, which is small enough to avoid deforming the simulated curve. The accuracies of both algorithms are not as high as in the case of an optimal scale (Figure S1), but our approach still achieves $\sim$85% of accuracy at $\sigma$ = 5 (medium noise) and $\sim$80% of accuracy at $\sigma$ = 10 (high noise) for both 1D, 2D and 3D, compared to much lower accuracies for the baseline in 2D and 3D. **Fig D: Intrinsic dimensional decomposition of the axonal trace of a retinal ganglion cell [40] across multiple scales.** Positions on the 3D trace are indexed by the curvilinear distance $u$ (x axis) and the scaled trace is calculated by Gaussian convolution with various standard deviations $s$ (y axis). An example of trace seen at different scales and superimposed with its decomposition is shown on the right. The trace exhibits mostly 3D at small scales, then decomposes into a combination of 1D/2D/3D portions at higher scales and finally transforms into a 1D line at a very high scale. The local 3D scale at each position $u$ is then measured as the minimal scale $s$ from that the dimension at $u$ is not 3D any more. **Fig E: Effect of sampling on local 3D scale computation.** Given a neuronal arbor (see Fig 2 for details) whose original sampling is around 6μm, the local 3D scale was compued for several different value of over and under sampling. While we clearly see a difference, the overall behaviour is robust to changes in sampling rate. **Fig F: Computation of local 3D scales in different decomposition modes on the cerebellar Purkinje neuron shown in Fig 2A3.** (Left) neurite portions located between any two branching nodes were extracted. (Middle) the longest branch originating from the cell body (root tree) was first extracted, and the process repeated for all subtrees extracted from that longest branch. (Right) branches connecting the cell body to each dendrite termination ('leaf') were extracted. The mean and standard deviation of the local 3D scale was computed. The "leaves" mode (right) produces more stable local 3D scales with high and homogenous values in region (i) where the dendrites are sticking out of plane compared to the "branching nodes" (left) and "longest branch" modes (middle) (See S3 Movie). **Fig G: Comparison between different local metrics of geometrical complexity.** Three parameters, curvature (left), torsion (middle) and local 3D scale (right) were mapped on the cerebellar Purkinje neuron shown in Fig 2A3. The local 3D scale gives smoother values than those of curvature and torsion since it was computed using a scale space approach, and better contrasts different regions of the Purkinje cell's dendritic arbor. **Fig H: Local 3D scales mapping across the whole larval zebrafish brain.** The traces analyzed correspond to those presented in [12]. (A) Coronal (top) and sagittal (bottom) views shiwing local 3D scale analysis of all axonal traces originating from the left hemisphere. (B) Mean local 3D scale by brain regions (values were clipped from $5^{th}$ to $95^{th}$ percentiles for clearer display). A transparency effect was applied to help visualizing inter-regional variations. **Fig I: Whole brain local 3D scale analysis of axons originating from different region of the larval zebrafish brain.** Traces analyzed correspond to those presented in [12]. (A) Mean local 3D scale by brain region (values were clipped in the same range as in Figure S6B for comparison). (B) Distribution of the local 3D scale values in each brain region. (C) Mean local 3D scale in fore- mid- and hindbrain. A Wilcoxon test with Holm-Sidak for multiple comparison was used, ns = not significant. (D, E) Correlation between the mean local 3D scales and average number of branching points (D) or trace length (E) in different brain regions. Spearman correlation was used. **Fig J: Whole brain local 3D scale analysis of axons passing through different region of the larval**

**zebrafish brain.** Traces analyzed correspond to those presented in [12]. (A) Mean local 3D scale by brain region (values were clipped in the same range as in Figure S6B for comparison). (B) Distribution of the local 3D scale values in each brain region. (C) Mean local 3D scale in fore- mid- and hindbrain. A Wilcoxon test with Holm-Sidak for multiple comparison was used, ns = not signficant. (D, E) Correlation between the mean local 3D scales and average number of branching points (D) or trace length (E) in different brain regions. Spearman correlation was used. **Fig K: Whole brain local 3D scale analysis of axons terminating in different region of the larval zebrafish brain.** Traces analyzed correspond to those presented in [12]. (A) Mean local 3D scale by brain region (values were clipped in the same range as in Figure S6B for comparison). (B) Distribution of the local 3D scale values in each brain region. (C) Mean local 3D scale in fore- mid- and hindbrain. A Wilcoxon test with Holm-Sidak for multiple comparison was used, ns = not signficant. (D, E) Correlation between the mean local 3D scales and average number of branching points (D) or trace length (E) in different brain regions. Spearman correlation was used. **Fig L: Variability of local 3D scale for axons originating from, passing through or arriving in three brain regions (MOS3, SP, TS).** The regional local 3D scale differs the three axons subset. **Fig M: Local 3D scale of axons crossing the midline.** (A) Distribution of the number of midline crosses made by individual axon arbors. (B) Distribution of the number of 'zigzags' through the midline, i.e. total number of crossings minus the number of crossing branches, for axons crossing more than 2 times. (C) Relation between the local 3D scale and length of crossing branches at the midline. A branch is defined as a neurite segment located between two branching points, or a branching point and a leaf or the root of the arbor. **Algorithm A:** A pseudo-code explanation of the nAdder computation algorithm. **Table A: Summary of 36 annotated brain regions from** [12]. **Table B: List of regions having axons originating from and arriving to the Torus Semicircularis (TS) from** [12]. Index 0 corresponds to axons not starting from any regions.
(PDF)

**S1 Movie. Visualisation in 3D of a mouse striatal D2-type medium spiny neuron reconstructed by [39].**
(MP4)

**S2 Movie. Visualisation in 3D of a mouse retinal ganglion cell [40].**
(MP4)

**S3 Movie. Visualisation in 3D of a mouse cerebellar Purkinje neuron from [41].**
(MP4)

## Author Contributions

**Conceptualization:** Katherine Matho, Jean Livet, Anatole Chessel.

**Data curation:** Katherine Matho, Jean Livet, Anatole Chessel.

**Formal analysis:** Anatole Chessel.

**Funding acquisition:** Emmanuel Beaurepaire.

**Investigation:** Minh Son Phan, Anatole Chessel.

**Methodology:** Minh Son Phan, Anatole Chessel.

**Software:** Minh Son Phan, Anatole Chessel.

**Supervision:** Emmanuel Beaurepaire, Jean Livet, Anatole Chessel.

**Validation:** Minh Son Phan.

**Visualization:** Minh Son Phan.

**Writing – original draft:** Minh Son Phan, Anatole Chessel.

**Writing – review & editing:** Katherine Matho, Emmanuel Beaurepaire, Jean Livet.

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
