## [Decision Letter · Decision Letter 0]

21 Mar 2022

Dear Dr Chessel,

Thank you very much for submitting your manuscript "nAdder: A scale-space approach for the 3D analysis of neuronal traces" for consideration at PLOS Computational Biology. As with all papers reviewed by the journal, your manuscript was reviewed by members of the editorial board and by several independent reviewers. The reviewers appreciated the attention to an important topic. Based on the reviews, we are likely to accept this manuscript for publication, providing that you modify the manuscript according to the review recommendations.

Please pay particular attention to the concerns raised by Reviewer 2. As they do not require additional experiments, this can be done in a minor revision, but must nevertheless be addressed thoroughly.

Sincerely,

Virginie Uhlmann

Associate Editor

PLOS Computational Biology

Daniele Marinazzo

Deputy Editor

PLOS Computational Biology

[LINK]

Please pay particular attention to the concerns raised by Reviewer 2. As they do not require additional experiments, this can be done in a minor revision, but must nevertheless be addressed thoroughly.

Reviewer's Responses to Questions

**Comments to the Authors:**

Reviewer #1: # General

Disclaimer: I am neither an expert in geometry nor in neuro-science, and I may thus overlook aspects that experts in those fields may spot.

For my point of view (possibly representing a general reader with a background in biophysics) I would like to congratulate the authors for this very well written manuscript! The presented method strikes me the same time as understandable, feasible and biologically meaningful! I'd anticipate a broad adoption of the method in the field of neuron trace inspection and classification.

I only have a few suggestions for improvements of the manuscript, figures, and software documentation.

# Manuscript text

Line 8: Could you add the gitlab link to the software to the abstract to facilitate uptake of the software?!

Line 106: I know one could look this up in the original publication, but would it be possible to add two sentences explaining how the convolution of a trace works? I assume one loops through the coordinates and replaces each coordinate with a weighted mean of the coordinates in the vicinity?!

Line 284: You speak about "fragments", could you please add an explanation to the manuscript what a "fragment" is? Is it something like "branches" (a term used in skeletonization of images)?

Line 352: "...as a new local metric...". Could you please add some discussion to the manuscript (maybe in the methods section) as to "how local" this really is? Since you need up to the 3rd derivative to compute the scale I guess you need at least some neighboring points on the trace? How much distance does this correspond to in micrometer units? Does this depend on the sampling of the trace? Related to that, do the traces need to be sampled equidistant for the method to work?

# Manuscript figures

## Legend for 1D,2D,3D

Could you please make the coloring coding for 1D,2D,3D consistent throughout the manuscript.

- Fig 1A has colors with dashes

- Fig 1B middle panel, is really hard to see (at the small print size), are there some yellow boundaries around 1D? Could they be removed to make it easier to look at?

- Fig S2 simple colors

- Fig S3 same as 1A

- Fig S4 there is some checkerboard pattern for 1D

I'd recommend a simple color-blind safe scheme without any borders or dashes or alike,

e.g. the scheme in Fig S2 would work for me.

## Scale bars

Please add scale bars such that one can see how large the trace is. Scale bars are missing in Figures: 1B, S2, S5, S6

# Software

## Notebook link broken

This notebook can be download from: https://gitlab.com/genepy3d/genepy3d/-/blob/master/docs/source/working_with_curve.ipynb

Yields: "docs/source/working_with_curve.ipynb" did not exist on "master"

Reviewer #2: The authors are introducing a novel local descriptor for space-curves. They are computing this new descriptor in existing datasets of neuronal arbors, and show that it is complementary to previously used morphological measurements of neuronal traces. Overall, the authors show that you can use this 3D scale metric for a fine neuronal trace description, which can bring more insights when comparing experimental conditions. This could be useful for a variety of neuroscience fields, and the applied data looks convincing.

However, after going through the paper several times, I still can’t get a clear understanding of what this local indicator is measuring. I don’t know how to interpret it and I wonder how the audience of neuroscientists will be willing to use a descriptor which looks quite complicated to understand.

I explain below a few points which mislead me and which could also mislead readers. I hope that my misdirections can be used by the authors to rewrite some parts of the paper to make it more easily understandable.

1 - Intuitively, when looking at a curve at a small scale, a 1D description is sufficient, because the curve follows its tangent. Then, looking a bit further away (higher up in the scales), one could use a plane (2D) to account for the curvature. Finally, at a higher scale, torsion has to be taken into account: the 3rd dimension becomes necessary to describe the curve. Following my wrong ‘intuition’, I was expecting that a 3D description is required at high scales, and a 1D description would be sufficient for low scales. But this intuition is wrong because Fig. 1B gives the inverse relation (small scale (15): 3D, large scale (150): 1D).

What I did not initially get is that the scale shown in the Y axis of Fig1B represents the degree of smoothing of the curve. Thus when the curve is heavily smoothed (150 microns), it can be ‘modeled’ as a line. Conversely, when it is less smoothed, it needs to be described within a 3D space. Ok.

However I do not really understand how you can derive a local property out of this observation. To get a local property, I believe that a FIXED ‘typical sampling size’ (let’s say 5 microns for instance) has to be introduced, independent from the smoothing scale. This sampling size would be the one to which you are comparing your smoothed curve at a particular scale. If no ‘typical sampling size’ is introduced, then looking with a 50 micron precision at a 50 micron-smoothed curve will give more or less the same ‘dimensionality’ as looking at a 10-micron smoothed curve with a 10 micron resolution.

So: shouldn’t there be an independent sampling size parameter for which the curve is analyzed? If there is an implicit one, where does it come from?

2 - In the abstract:

“we propose a local tridimensional (3D) scale metric derived from differential geometry, defined as the distance in micrometers along which a curve is fully 3D as opposed to being embedded in a 2D plane or 1D line”.

What’s surprising is that ‘locally’ a differentiable curve is a line, thus 1D. If you sample the curve over a finite portion, at a certain size, then I understand that you may consider the curve being 1D or 2D or 3D over this portion. But there is no mention of a ‘typical size’ in the abstract, thus leading to my confusion. Is there any way this sentence can be rewritten to understand better the distance over which the curve is considered 3D?

3 - In materials and methods:

I think part of the answers to my questions about the ‘typical size’ is answered in L270 to L283. But here I do not really understand the logic for the complicated scaling. Why not always choose a fixed set of scales with a standard set of sizes (2 microns, 4 microns, 8 microns, up to infinity)? Wouldn’t that be easier to understand ? Would that impact the results ?

Overall, I believe that a longer and broader method section that precisely goes through all steps of the math is required to make the metric easier to understand. Many synthetic simple example curves throughout this section could lead to a better understanding of the method. (broken lines, spirals, lines intertwined with circles, etc.).

There is one example in supp Fig.1, but I do not know how the scale decomposition shown in Fig1B would look like with this the supp Fig. 1 example. Also, how the supp Fig S4 would look if color coded like the Fig 1B (right) ?

Some other remarks:

It is unclear to me what the graphs in figure 1A (right) are representing, and the caption is not helpful.

L274 “For a given curve, the level of detail can be characterized by the maximal rκ along that curve.” -> unclear which level of detail ? I guess the lowest level of detail. But isn’t the lowest level of detail just a line joining both ends ( with an infinite rk ) ?

Typos:

L263 diferentiable -> differentiable

L276 keeing -> keeping

Reviewer #3: The authors provide a novel metric to parametrize a 3D curve. They call it Neurite Analysis through Dimensional Decomposition in Elementary Regions or ‘nAdder’. The metric is based on estimating the local 3D scale along a curve.

The concept of the proposed metric is well presented in Fig 1. Then, the authors highlight the usefulness of the proposed metric to characterize differences in previously traced neurons where traditional metrics failed to revealed any significant difference (Fig. 2B). To showcase the ability of the new metric to generate insights within larger datasets authors use two examples, the whole larval zebrafish brain (Fig. 3) and the axonal behavior along defined anatomical locations and along specific tracts. Both examples show clearly the utility of the proposed metric in characterizing neuronal populations and brain regions.

Overall, the paper in its current form is well-written, clear, with carefully selected examples where the new metric generates new insights and is well-discussed. I see no need to add any further comments or request additional material.

I would love to see in the future how this new metric can be paired with molecular profiling experiments to compare geometric phenotype with the proteomic or transcriptomic profiling of individual neurons.

A single minor suggestion, to the sentence of p4 line 112: “we define the local 3D scale as the highest scale at which the trace still remains 3D” somehow the concept that is a local measure is not included. Could this be rephrased to include that information?

**Have the authors made all data and (if applicable) computational code underlying the findings in their manuscript fully available?**

Reviewer #1: Yes

Reviewer #2: Yes

Reviewer #3: Yes

PLOS authors have the option to publish the peer review history of their article (what does this mean?). If published, this will include your full peer review and any attached files.

Reviewer #1: No

Reviewer #2: No

Reviewer #3: No

Figure Files:

Data Requirements:

Reproducibility:

References:

---

## [Editor Report · Decision Letter 1]

16 May 2022

Dear Dr Chessel,

We are pleased to inform you that your manuscript 'nAdder: A scale-space approach for the 3D analysis of neuronal traces' has been provisionally accepted for publication in PLOS Computational Biology.

Best regards,

Virginie Uhlmann

Associate Editor

PLOS Computational Biology

Daniele Marinazzo

Deputy Editor

PLOS Computational Biology

---

## [Editor Report · Acceptance letter]

27 Jun 2022

PCOMPBIOL-D-22-00209R1 

nAdder: A scale-space approach for the 3D analysis of neuronal traces

Dear Dr Chessel,

I am pleased to inform you that your manuscript has been formally accepted for publication in PLOS Computational Biology. Your manuscript is now with our production department and you will be notified of the publication date in due course.

With kind regards,

Zsofi Zombor
